# Astrocyte-Derived Small Extracellular Vesicles Regulate Dendritic Complexity through miR-26a-5p Activity

**DOI:** 10.3390/cells9040930

**Published:** 2020-04-10

**Authors:** Alejandro Luarte, Roberto Henzi, Anllely Fernández, Diego Gaete, Pablo Cisternas, Matias Pizarro, Luis Federico Batiz, Isabel Villalobos, Matias Masalleras, Rodrigo Vergara, Manuel Varas-Godoy, Lorena Abarzua-Catalan, Rodrigo Herrera-Molina, Carlos Lafourcade, Ursula Wyneken

**Affiliations:** 1Centro de Investigación e Innovación Biomédica (CIIB), Facultad de Medicina, Universidad de los Andes, Santiago 7550000, Chile; rpjhenzi@gmail.com (R.H.); anllely.fernandez@gmail.com (A.F.); diegogaete@gmail.com (D.G.); pcistern@gmail.com (P.C.); matias.pizarroc@mayor.cl (M.P.); LBATIZ@uandes.cl (L.F.B.); isavillalobos93@gmail.com (I.V.); mmasalleras@gmail.com (M.M.); lorena.abarzua@gmail.com (L.A.-C.); clafourcade@gmail.com (C.L.); 2Biomedical Neuroscience Institute, Universidad de Chile, Santiago 8380453, Chile; rvergara@bni.cl; 3Departamento de Kinesiología, Facultad de Artes y Educación Física, Universidad Metropolitana de Ciencias de la educación, Santiago 7780450, Chile; 4Cancer Cell Biology Lab, Centro de Biología Celular y Biomedicina (CEBICEM), Facultad de Medicina y Ciencia, Universidad San Sebastián, Santiago 7510157, Chile; manuel.varas@uss.cl; 5Synaptic Signaling Lab, Leibniz İnstitute for Neurobiology; Center for Behavioral Brain Sciences, 39118 Magdeburg, Germany; rherrera@lin-magdeburg.de; 6Centro Integrativo de Biología y Química Aplicada, Universidad Bernardo O’Higgins, Santiago 8370993, Chile

**Keywords:** microRNAs, exosomes, astrocytes, hippocampal neurons, dendritic complexity

## Abstract

In the last few decades, it has been established that astrocytes play key roles in the regulation of neuronal morphology. However, the contribution of astrocyte-derived small extracellular vesicles (sEVs) to morphological differentiation of neurons has only recently been addressed. Here, we showed that cultured astrocytes expressing a GFP-tagged version of the stress-regulated astrocytic enzyme Aldolase C (Aldo C-GFP) release small extracellular vesicles (sEVs) that are transferred into cultured hippocampal neurons. Surprisingly, Aldo C-GFP-containing sEVs (Aldo C-GFP sEVs) displayed an exacerbated capacity to reduce the dendritic complexity in developing hippocampal neurons compared to sEVs derived from control (i.e., GFP-expressing) astrocytes. Using bioinformatics and biochemical tools, we found that the total content of overexpressed Aldo C-GFP correlates with an increased content of endogenous miRNA-26a-5p in both total astrocyte homogenates and sEVs. Notably, neurons magnetofected with a nucleotide sequence that mimics endogenous miRNA-26a-5p (mimic 26a-5p) not only decreased the levels of neuronal proteins associated to morphogenesis regulation, but also reproduced morphological changes induced by Aldo-C-GFP sEVs. Furthermore, neurons magnetofected with a sequence targeting miRNA-26a-5p (antago 26a-5p) were largely resistant to Aldo C-GFP sEVs. Our results support a novel and complex level of astrocyte-to-neuron communication mediated by astrocyte-derived sEVs and the activity of their miRNA content.

## 1. Introduction

Astrocyte clues such as secreted factors (secretome) and cell–cell contact signals are essential for proper development, maintenance, and functioning of individual neurons, as well as for the wiring of the central nervous system (CNS) by controlling axonal guidance and dendritic complexity [1,2,3].

Several studies have addressed the molecular mechanisms mediating the role of the astrocyte-derived secretome on dendritic morphology. To name some of these factors, it has been shown that astrocytes release phosphatidic acid to promote dendritic complexity in cultured hippocampal neurons as well as apolipoprotein E-complexed cholesterol and hevin to increase synaptogenesis in cultured retinal ganglion cells [4,5,6]. Very recently, it has also been shown that astrocyte-derived small extracellular vesicles (sEVs) act as regulators of cell functions and signaling in CNS cells, especially between astrocytes and neurons [7,8,9]. The most studied types of sEVs, also known as exosomes, are vesicles of 30–120 nm in diameter with an endocytic origin. These nanosized sEVs are released to the extracellular space from multivesicular bodies (MVBs) after their fusion with the plasma membrane [10]. It has been claimed that sEVs’ molecular cargo could modify the physiology of recipient cells through the transfer of active micro RNAs (miRNAs) [10,11,12,13]. For instance, mature miRNAs, i.e., small non-coding RNAs, 20–22 nucleotides long, recognize specific sequences located mainly at the 3` untranslated region of mRNA transcripts and, thus, they can determine either their translational quiescence or downregulation. The mechanisms involved in regulating miRNA loading in sEVs as well as the roles of miRNAs contained in astrocyte-derived sEVs remain mostly unexplored [14,15,16].

We have shown that the astrocyte’s specific glycolytic enzyme Aldolase C (Aldo C) is present in sEVs and that its content in the vesicles is regulated in vivo and in vitro: Astrocyte-derived sEVs contain Aldo C and its levels increase in rat cerebrospinal fluid and in serum sEVs after exposure of animals to stress induced by movement restriction [17,18,19]. In addition, we have recently shown that Aldo C expressed in brain astrocytes can be collected in sEVs isolated from rat blood serum, supporting the capacity of astrocyte-derived sEVs to cross biological barriers and to serve as regulators of intercellular signaling [19]. However, it is not known whether neurons, which are in close proximity to astrocytes, can incorporate the derived sEVs containing Aldo C nor whether they could impact neuronal function.

Here, we showed that cultured astrocytes that express GFP-tagged Aldo C (Aldo C-GFP) transfer the derived sEVs, carrying the recombinant protein to developing hippocampal neurons, impacting their dendritic complexity. Using bioinformatics combined with biochemical and molecular approaches, we postulated and then confirmed that the content of miRNA-26a-5p is regulated in Aldo C-GFP-electroporated astrocytes and their sEVs. Finally, we showed that the miRNA-26a-5p carried by Aldo C-GFP-containing sEVs (Aldo C-GFP sEVs) actively regulates the expression of some neuronal proteins that are relevant in morphogenesis and the regulation of dendritic complexity in a manner dependent on the activity of miRNA-26a-5p.

## 2. Materials and Methods

### 2.1. Animal Procedures

For all described protocols, pregnant Sprague Dawley rats were used at E18 following ethical guidelines approved by the Universidad de Los Andes Bioethical Committee associated to Fondecyt project number: 1140,108 (Protocol # 09112013) and in accordance with the National Institute of Health’s Guide for the Care and Use of Laboratory Animals. All techniques were performed with all efforts to minimize animal suffering.

### 2.2. Plasmids

To obtain the stable expression of transgenic proteins in astrocytes, we used a system consisting of a donor and helper plasmid (piggyBac), as published elsewhere [20]. All reading frames were under the control of the ubiquitous and strong cytomegalovirus early enhancer/chicken β actin promoter (CAG) previously described in [21]. To construct the donor, plasmid rat gene sequence coding for Aldo C or GFP proteins were cloned into pPBCAG_eGFP plasmids using the restriction sites of EcoRI and AgeI. pPBCAG-Pbase plasmids were used as helpers. Both backbone constructs were kindly donated by LoTurco [20].

### 2.3. Primary Antibodies and Dilutions

The primary antibodies dilutions in this study were used at 1:1000, except when indicated. They were: MAP2 A/B (MAB5622, Millipore, Billerica, MA, United States); MAP2 A/B (MAB378, Millipore, Billerica, MA, United States); Aldo C (sc-12065, Santa Cruz Biotechnology, Dallas, TX, United States); TSG101 (Ab83, Abcam, Cambridge, MA, United States); Flotilin-1 (610821, BD, Franklin Lakes, NJ, United States); CD63 (sc-15363, Santa Cruz Biotechnology, Dallas, TX, United States); GM130 (Ab52649, Abcam, Cambridge, MA, United States); GFAP ( G2032-28B-PE, US Biological, Salem, MA, United States); GFP (Ab6673, Abcam, Cambridge, MA, United States); GFP (MAB3580, Millipore, Billerica, MA, United States) (From our lab. This antibody detects a faint band over 63 kDa in astrocyte homogenates); Alix (sc-53540, Santa Cruz Biotechnology, Dallas, TX, United States); and β-actin (A5441, Sigma-Aldrich, St Louis, MO, United States).

### 2.4. Secondary Antibodies and Dilutions

The following secondary antibodies were used at 1:1000 dilutions in immunofluorescence and 1:5000 dilutions for Western blots: Alexa Fluor^®^ 488 donkey anti mouse IgG (H+L) (a21202, Thermo Fisher Scientific, Waltham, MA, United States); Alexa Fluor^®^ 555 goat anti rabbit IgG (H+L) (a21429, Thermo Fisher Scientific, Waltham, MA, United States); Alexa Fluor^®^ 488 goat anti rabbit IgG (H+L) (a11034, Thermo Fisher Scientific, Waltham, MA, United States); anti-mouse rabbit anti-IgG horseradish peroxidase conjugated antibody (# 31430, Thermo Fisher Scientific, Waltham, MA, United States); anti-rabbit IgG horseradish peroxidase conjugated antibody (# 31460, Thermo Fisher Scientific, Waltham, MA, United States); and anti goat IgG horseradish peroxidase conjugated antibody (# 31402,Thermo Fisher Scientific, Waltham, MA, United States).

### 2.5. Immunofluorescence (IF)

Cultured neurons and astrocytes were fixed with 100% *w*/*v* methanol at −20 °C for 5 min, further permeabilized with 0.2% *w*/*v* Triton X-100 in phosphate-buffered saline (PBS) for 5 min and blocked with 10% *w*/*v* BSA in PBS for 10 min. Then, the cells were incubated overnight at 4 °C with the corresponding primary antibody diluted in 10% *w*/*v* BSA in PBS. Then, cells were washed 3 times with PBS (5 min each) and incubated at room temperature for 1 h with the corresponding secondary antibody coupled to a fluorescent dye. Subsequently, the cells were washed 3 times (for 5 min) and incubated with 300 nM 4ʹ, 6-diamidino-2-phenylindole (DAPI) in PBS for 3 min. Finally, the cells were mounted using the fluorescence-mounting medium (DAKO, Hamburg, Germany). The samples were analyzed in a NIKON TE-2000U epifluorescence microscope (Nikon Instruments Inc, Melville, NY, United States) equipped with a DS-2MBWC camera (2.0 monochromatic CCD megapixels). In addition, confocal microscopy was performed in an Olympus FluoView FV1000 device (Olympus, Shinjuku, Tokyo, Japan)with a UPLSAPO 60×/1.35 objective. Some samples were analyzed under Leica SP8 confocal microscope (Leica, Wetzlar, Germany).

### 2.6. Western Blot

For protein extraction, cells were washed twice with cold PBS and lysed with cold RIPA buffer (50 mM Tris-HCl (pH 7.4),150 mM NaCl, 0.25% deoxycholic acid, 1% NP-40, 1% SDS, and 1 mM EDTA). Protein concentration was measured using the bicinchonic acid method (BCA), according to the Pierce BCA Protein Assay Kit (Thermo Fisher Scientific, Waltham, MA, United States).

Proteins were separated using sodium dodecyl sulfate-polyacrylamide gels (SDS-PAGE) under fully denaturing conditions. Electrophoresis was performed at 70 V for 45 min, finishing at 100 V in linear 12% p/v acrylamide gels. The transfer of proteins from the gel to a nitrocellulose membrane (Bio-Rad Laboratories, Hercules, CA, United States ) was performed using a constant current of 350 mA for 90 min. Then, membranes were blocked with 5% *w*/*v* skim milk in PBS for 1 h at room temperature under constant agitation. Membranes were washed 3 times for 5 min with PBS to remove the excess milk and incubated at 4 °C with the corresponding antibody diluted in PBS with constant shaking overnight. Membranes were then washed 3 times with 0.1% *w*/*v* Tween in PBS for 10 min and incubated with the corresponding secondary antibody in a 1:5000 dilution with 0.1% *w*/*v* Tween in PBS and 5% p/v skim milk for 1 h at room temperature. Membranes were washed 2 times with 0.1% *w*/*v* Tween in PBS for 10 min and once with PBS. Finally, membranes were incubated for 1 min with the chemiluminescent reagent (Amersham Bioscience, Piscataway, NJ, United States) and then exposed to the film (Hyperfilm, ECL, Amersham Bioscience, Piscataway, NJ, United States). Bands were quantified by densitometry using the Adobe Photoshop 7.0 software (Adobe Inc., San José, CA, United States).

### 2.7. RNA Extraction

Isolated sEVs and cell cultures were processed with the miRNeasy Plus Mini Kit (Qiagen, Hilden, Germany), according to the manufacturer’s instructions. The starting material was quantified as the total amount of proteins: 400 μg for cells and 10 μg for sEVs were used for each experimental condition. The samples were quantified using a microvolume spectrophotometer Nanodrop 2000 (Thermo Fisher Scientific, Waltham, MA, United States). The concentration was determined with the absorbance at 260 nm (A260), while the purity was estimated by measuring the absorbance ratio 260/280.

### 2.8. Reverse Transcription Quantitative PCR

The reverse transcription to obtain the cDNAs was done with the TaqMan^®^ MicroRNA Assays (Roche, Basilea, Switzerland) commercial kit, according to the manufacturer’s instructions. For each experimental condition in all the experiments of this publication, 100 ng of total RNA were mixed with primers of miR-26a-5p and miR-26a-3p or U6 (a component of the splicing machinery), plus the Mixing buffer for retro-transcription. The reaction was performed in a thermocycler with the following temperature program: 30 min at 16 °C; 30 min at 42 °C; and 5 min at 85 °C. The reaction was stopped at 4 °C to then perform a reverse transcription quantitative PCR (RT-qPCR) according to the manufacturer’s instructions. Briefly, specific Taqman^®^ primers were mixed together with the cDNAs, plus a solution composed of RNAase free water and Taqman universal master mix II. Each one of those mixes was submitted to the following temperature cycle program: an initial step to activate DNA polymerase for 10 min at 95 °C, followed by 40 cycles of 15 s at 95 °C; and 60 s at 60 °C to allow amplification. Once the amplification cycles were completed, we obtained the value of cycle threshold (Ct). The U6-corrected fold change of miRNA content was calculated using the procedure described in [22].

### 2.9. In Utero Electroporation

In utero electroporation was performed to stably express proteins in astrocytes that can be maintained in culture in excellent conditions for at least 3 weeks. Electroporation of rat cerebral cortices was done in E18-E19 embryos as previously described [23]. The fetuses were exposed through a 4 cm incision over the linea alba of the muscular layer. Once exposed, the fetuses were constantly moisturized with a saline solution (0.9% *w*/*v* NaCl in distilled water) plus antibiotics, using a dilution of 50 U/mL penicillin and 10,000 μg/mL streptomycin at 37 °C. Previously, we prepared glass capillaries with a 100–150 μm inner radius using a P97 pipette puller (Sutter Instruments, Novato, CA, United States) filled with 25 μL of a solution with the following composition: 0.75 μg/μL Fast Green dye (Sigma-Aldrich, St Louis, MO, United States); 1 μg/μL pPBCAG_Pbase plasmid, and 1 μg/μL pPBCAG_Aldo C-GFP or pPBCAG_ GFP plasmids, all diluted in distilled water. Then, 1–2 μL of this solution was injected into the left lateral ventricle of each embryo by means of a peak-pressure pump PV830 ( World Precision Instruments, Sarasota, FL, United States). An electric pulse of 60–70 Volts was given by a capacitor of 500 μF previously charged with a 250 V power source. The discharge was done through copper alloy plates (1 × 0.5 cm) and arranged over the brain with the positive electrode facing the left hemisphere. Fetuses were allowed to grow in utero until day 21 of gestation to perform pure astrocyte cultures. For this, the brain was observed under a stereo microscope with the help of a fluorescence adapter (NightSea, Lexington, MA, United States) to select tissue with transgene expression from the electroporated left telencephalon.

### 2.10. Cell Cultures and Isolation of sEVs

Hippocampal neurons were obtained from embryonic Sprague Dawley rats (E18) as previously described [24]. Primary astrocyte cultures (90% of GFAP-positive cells) were obtained following established procedures with slight modifications [25]. In order to clearly identify fetuses electroporated with different plasmids, we started astrocyte cultures at E21. When cultured astrocytes reached 70–90% confluence, culture media was replaced by a sEV free medium for 72 h. Subsequently, the medium was collected, and successive centrifugations were performed: 30 min at 2000× *g* to eliminate cells and debris; 40 min at 10,000× *g* to eliminate microvesicles; and 2 h at 100,000× *g* to obtain the sEV enriched fraction in the pellet. Finally, this pellet was washed by resuspension in phosphate buffer saline (PBS) at pH = 7.4 and centrifuged again for 2 h at 100,000× *g* to obtain the 100 K pellet fraction. The resulting pellet is enriched in sEVs [26].

### 2.11. Transwell Astrocyte-Neuron Co-Culture

In order to obtain astrocytes expressing Aldo C-GFP or GFP, in-utero electroporation was performed as described above and astrocyte primary cultures were performed. After 15 days in vitro (DIV), the Aldo C-GFP or GFP-electroporated astrocytes were treated with trypsin (Sigma-Aldrich, St Louis, MO, United States) for 1 min at 37 °C, and, finally, the cells were seeded in a polycarbonate Transwell system of 0.4 µm pores (Corning Costar Co., Cambridge, MA, United States ) to reach 70% confluency. The Aldo-GFP or GFP astrocytes were transferred 24 h later to 24 plate wells to be co-cultured with 3 DIV hippocampal neurons. Three days later, both astrocytes and neurons were fixed using 4% PFA, and samples were submitted to IF.

### 2.12. Nanoparticle Tracking Analysis (NTA)

The sEVs were analyzed with the NanoSight LM-10 device (Malvern Instruments, Malvern, UK) equipped with a green laser, as described in [19].

### 2.13. Sucrose Flotation Assay

To perform the assay, 300–400 µg of sEVs were resuspended in 1 mL of 2.5 M sucrose with 50mM HEPES buffer at pH = 7.2, diluted in deuterated water, and loaded at the bottom of a 13 mL ultracentrifuge tube. A continuous linear gradient was made from 2 M and 0.5 M sucrose solutions prepared with 50 mM HEPES buffer at pH = 7.2, diluted in deuterated water, and added over exosomes. The gradient was centrifuged for 17 h at 200,000× *g* and stopped with the free braking mode of the ultracentrifuge. Then, 1 mL fractions were collected from the top of the tube with the sucrose gradient. Each of these fractions was resuspended in 12 mL of 50 mM HEPES at pH = 7.4, and then centrifuged at 200,000× *g* for 2 h in order to collect sEVS in the precipitate. Each pellet was resuspended in 30 μL of loading buffer and boiled under denaturant conditions and fully loaded on a 12% *w*/*v* acrylamide gel with SDS for analysis by Western blot.

### 2.14. Incubation with sEVs

Isolated sEVs were resuspended in Neurobasal medium and added onto 20,000 hippocampal neurons (3 DIV) to obtain a final protein concentration of 10 ng/μL, in a total volume of 400 μL. As assessed by NTA analysis, this corresponds to a total mean number of added vesicles of 0.95 × 10^9^ particles for Aldo C-GFP sEVs and 1.2 × 10^9^ particles for GFP sEVs. After 72 h, neurons were fixed, stained with the corresponding antibodies, and submitted to Sholl analysis. When indicated, neurons were treated with sEVs 2 h after magnetofection. For the uptake experiments of sEVs, 200,000 hippocampal neurons were seeded on 35 mm plates or 25 mm coverslips and at 6 DIV they were incubated with 10 µg (protein content) of the corresponding sEVs at 4 °C or 37 °C for three hours. For Western blot analysis, the complete cell lysate obtained from each well/condition was loaded in each lane. For IF analysis, cells were fixed with 4% PFA. Alternatively, cells were lysed for RNA extraction and submitted to quantitative RT-PCR.

### 2.15. Neuronal Magnetofection

For neuronal transfections, 10 pmol of miR-26a-5p mimic (mimic 26a-5p) (Ambion^®^ # 4464066, Thermo Fisher Scientific, Waltham, MA, United States), miR-26a-5p inhibitor or antago (antago 26a-5p)( Ambion^®^ # 4464084, Thermo Fisher Scientific, Waltham, MA, United States), and miR-26a-5p mimic negative control (scrambled) (Ambion^®^ # 4464058, Thermo Fisher Scientific, Waltham, MA, United States ) were transfected by NeuroMag Transfection Reagent (Ozbioscience, San Diego, CA, United States) following the manufacturer’s instructions. High transfection efficiency was achieved by magnetofecting small fluorescent oligonucleotides (>90%, data not shown). Similar results were obtained in the literature [27]. When indicated, sEVs were added 2 h after neurons were magnetofected with the respective oligos. Additionally, 200,000 neurons seeded on 35 mm plates were magnetofected with 20 pmol of the respective oligos and submitted to Western blot.

### 2.16. Morphological Analysis

Neurons were submitted to a Sholl analysis using the plugin of the Image J software [28]. All concentric radiui were 3 μm from each other. The following parameters were obtained: total intersections (i.e., the sum of all intersections with each different radius); primary intersections (i.e., number of intersections with the first radius); critical distance (i.e., the radius with the maximum number of intersections); maximum number of intersections (i.e., maximum number of intersections reached by a neuron at any radius); maximum distance (i.e., the largest radius at which there is an intersection with a neuronal process).

### 2.17. Bioinformatic Analysis

The miRECORDS platform (http://tinyurl.com/js9jr8n) was used to identify theoretical targets of miRNAs enriched in astrocytes by the consolidation of eleven programs with different prediction algorithms: DIANA-microT, MicroInspector, miRanda, MirTarget2, miTarget, NBmiRTar, PicTar, PITA, RNA22, and RNAhybrid and TargetScan/TargertScanS. We selected the targeted genes predicted by at least four different algorithms. The obtained list was further analyzed with the functional enrichment tool in biological processes defined by the AmiGO2 platform (http://tinyurl.com/z2pn5hb) [29,30,31]. To obtain a functional enrichment of the genes associated with cell functions, we normalized it by the expected value of the gene from a human reference genome. The list of predicted genes and the corresponding functional enrichment analysis for each miRNA are provided in the Appendix A.

### 2.18. Statistical Analysis

Differences between two groups were determined using the Welch’s *t*-test. The differences between more than two groups were evaluated with one-way ANOVA followed by Tukey’s post-hoc test. The fold changes compared to a theoretical value were evaluated with one-sample *t*-test or Wilcoxon one-sample signed-rank test when indicated. The differences were considered statistically significant with *p* < 0.05.

## 3. Results

### 3.1. Astrocytes Transfer Aldolase C-GFP-Containing sEVs to Neurons

In order to unequivocally identify sEVs derived from astrocytes, we electroporated in utero rat cerebral cortices with plasmids to overexpress Aldolase C-GFP (Aldo C-GFP) or GFP (GFP) as control, using the cytomegalovirus early enhancer/chicken β actin promoter (CAG). Then, the electroporated cerebral cortices were used to obtain primary astrocyte cultures (see Methods; Figure 1A). Expression of each of the transgenic constructs was observed in cultured astrocytes by Western blot (Figure 1B). Note that we detected a faint slight band over 63 kDa in all the lanes, which corresponded to unspecific staining for this particular GFP antibody (indicated in Materials and Methods) and not to the leak of transgenic constructs. To examine transference of sEVs from astrocytes to neurons, astrocytes were seeded in the upper chamber while 3 DIV hippocampal neurons were placed in the lower chamber of a Transwell co-culture system (Figure 1C). After 72 h of co-culturing, neurons were fixed, stained with anti-MAP2 antibodies to reveal their somatodendritic compartments, and analyzed under a confocal microscope. We observed abundant GFP-positive puncta (GFP+) in neurites and cell bodies of neurons co-cultured with Aldo C-GFP-expressing astrocytes, while virtually no GFP signal was found in neurons co-cultured with GFP-expressing astrocytes (Figure 1D). Notably, Aldo C-GFP nanometric puncta were found intracellularly in neurons, and in a fashion that suggests transference of Aldo C-GFP from astrocytes into neurons through extracellular vesicles.

Aldolase C is an astrocytic and cytoplasmic glycolytic enzyme released in sEVs/exosome-like fractions [17,18,19]. We characterized the sEV fractions from cultured astrocytes by ultracentrifugation and Western blot analysis. In the 100 K pellet fraction that contains sEVs, the proteins Alix, CD63, and TSG101 were detected (upper panel, Figure 1E), which are well accepted sEV markers of endosomal origin. As evidence of a selective enrichment of sEVs but not of other secretory organelles, the Golgi marker GM130 was not detected in the isolated sEV fractions. Endogenous Aldo C and the glial acidic fibrillary protein (GFAP) were clearly detected in the sEV fractions (upper panel, Figure 1E). Similarly, we collected a fraction of sEVs derived from Aldo C-GFP-expressing astrocytes and detected bands at 63 kDa with antibodies against GFP and Aldo C. Thus, an Aldo C-GFP-containing fraction of sEVs, or Aldo C-GFP sEVs, were obtained (lower panel, Figure 1E). An expanded Western blot of Figure 1E showed that Aldo C-GFP astrocytes generated two bands when stained with an Aldo C antibody (corresponding to the recombinant and endogenous proteins), while only one band (the recombinant) was detected when stained with a GFP antibody (Appendix A). Surprisingly, Aldo C detection in sEVs revealed consistently higher levels of the recombinant protein (near 63kDa). The endogenous protein only appeared after high over-exposure (not shown). Thus, Aldo C-GFP is loaded in sEVs more efficiently than the endogenous protein.

We further characterized the fraction containing astrocyte-derived sEVs by submitting the ultracentrifuged pellet to a flotation assay on a sucrose density gradient. In agreement with an endosomal origin of this fraction, Aldo C co-distributed in two peaks with the markers CD63 and TSG101. The first peak displayed a density around 1.16 g/mL, while the second had a peak density near 1.28 g/mL. These results are strongly compatible with previous work characterizing the density of sEV-containing fractions [10,32] (Appendix A). Then, we estimated the size of the astrocyte-derived sEVs by nanoparticle tracking analysis (NTA). In good agreement with the literature, the average size of the astrocyte-derived sEVs obtained with this methodology was of 155 ± 1 nm (Figure 1F). The average size was 173 ± 3 nm for Aldo C-GFP sEVs and 158.8 ± 2 nm for GFP sEVs. The total number of isolated vesicles remained unchanged among the three conditions (Appendix A).

To evaluate the incorporation of astrocyte-derived sEVs into neurons, 6 DIV hippocampal neurons were incubated with Aldo C-GFP sEVs at 4 °C or 37 °C for 3 h, and then washed, homogenized, and analyzed by Western blot (Figure 1G). A 63 kDa band was specifically detected with an anti-GFP antibody in homogenates of neurons incubated with Aldo C-GFP-containing sEVs at 37 °C, but not at 4 °C nor without added sEVs (vehicle) at 37 °C. When the localization of GFP was evaluated by immunocytochemistry and confocal microscopy, the large majority of GFP puncta was observed in neurons incubated with Aldo C-GFP sEVs at 37 °C. In contrast, the GFP signal was neglectable in neurons incubated with Aldo C-GFP sEVs at 4 °C (Figure 1H). Therefore, the internalization of astrocyte-derived sEVs by neurons seems to be an active and temperature-dependent process.

### 3.2. Astrocyte-Derived sEVs Decrease Dendritic Complexity in Neurons

We next evaluated whether astrocyte-derived sEVs regulate neuronal morphology. For this, 3 DIV hippocampal neurons were treated with astrocyte-derived sEVs for 72 h, stained with an anti-MAP2 antibody, and the dendritic complexity was quantified using Sholl analysis. Neurons treated with GFP sEVs displayed significantly decreased dendritic complexity compared to neurons under control conditions (left, Figure 2A). Indeed, the quantification of some morphometric parameters, including the number of intersecting dendrites (right, Figure 2A), and others, (Figure 2B, see Material and Methods) confirmed that sEVs can regulate the length and number of dendrites. More interestingly, Aldo C-GFP sEVs induced a larger decrease in all the morphometric parameters compared to GFP sEVs (Figure 2A,B). In the Transwell co-culture system, hippocampal neurons co-cultured with Aldo C-GFP-expressing astrocytes displayed a much simpler complexity compared to neurons co-cultured with GFP-expressing astrocytes (Figure 2C), as confirmed quantitatively (Figure 2C,D). These results indicate that sEVs derived from astrocytes regulate the morphological complexity of neurons and suggest that this regulation is influenced by the content of Aldo C. Nevertheless, the dendritic complexity of neurons directly transfected with Aldo C-GFP did not differ from control neurons (data not shown). Thus, it seems that molecular cargo different from Aldo C mediates the Aldo C-GFP sEVs-induced morphological changes in neurons.

### 3.3. Astrocyte-Derived sEVs Carry miR-26-5p, Which Targets Gene Expression Associated to Neuronal Development and Morphology and Regulates Protein Expression in Neurons

In order to identify molecular cargoes in astrocyte-derived sEVs able to promote morphological changes in neurons, we took advantage of a comprehensive study by Jovičić et al. identifying miRNAs in total homogenates from cultured astrocytes [33]. To generate a robust short-list of candidates, we selected the most abundant miRNAs (cycle threshold < 25), which have been confirmed to be present in astrocytes by at least one other independent study [13,34,35,36,37,38,39,40,41,42]. Then, the identified candidates, namely miR-26a5p, miR-23a-5p, miR-223a-5p, miR-19a-5p, miR-32a-5p, miR-146a-5p, miR-181-5p, and miR-29a-5p, were categorized using a selection algorithm according to the function of their predicted targets (see Methods; Figure 3A). Notably, the identified miRNAs shared a number of targets significantly associated with the regulation of cytoskeleton organization, neuronal development, Wnt signaling pathway, and morphogenesis of a branched structure (Figure 3A). In turn, the predicted targets for miR-26a-5p, miR-223a-5p, miR-19a-5p, miR-32a-5p, miR-29a-5p, and miR-181a-5p shared the more general category neuron development, while targets for miR-146a-5p and miR23a-5p showed no significant functional enrichment for any category (Figure 3A). Very interestingly, miR-26a-5p (miR-26a) was found to regulate targets participating in all four categories with a high functional enrichment score (Figure 3A). This miRNA was detected in control astrocyte-derived sEVs using RT-qPCR (cycle threshold = 30.3 ± 1.3, *N* = 6). Consistent with this finding, the presence of miR-26a-5p in astrocyte-derived exosomes has been described previously [33]. 

To validate miR-26a-5p activity in the regulation of neuronal morphology, we analyzed the protein levels of its validated target gene products microtubule associated protein 2 (MAP2) [43] and glycogen synthase kinase 3 β (GSK3β) [44], which also play key functions in neuronal morphology [45,46]. For this, we evaluated by Western blot the protein levels of MAP2 and GSK3β in hippocampal neurons magnetofected either with a scrambled control nucleotide sequence, a nucleotide sequence that mimics endogenous miRNA-26a-5p (mimic 26a-5p), or an inhibitor nucleotide sequence targeting miRNA-26a-5p (antago 26a-5p) for 72 h. As expected, the levels of MAP2 and GSK3β were reduced by mimic 26a-5p compared to scrambled or antago 26a-5p (Figure 3B). Then, the levels of miR-26a-5p in total homogenates and sEVs derived from GFP- and Aldo C-GFP-expressing astrocytes were evaluated by reverse transcription quantitative PCR (RT-qPCR) (left, Figure 3C). We found that miR-26a-5p in the homogenate of Aldo C-GFP-expressing astrocytes was 2.8 ± 0.9-fold higher than in GFP-expressing astrocytes. Surprisingly, increased miR-26a-5p levels in Aldo C-GFP sEVs were 31 ± 20-fold higher than in GFP sEVs (left, Figure 3C). The complete list of measured fold changes and their corresponding real time RT-PCR cycle threshold (Ct) values for both homogenates and sEVs are provided in the Appendix A.

Next, we wanted to determine if astrocyte-derived sEVs have the capacity to modify miR-26a-5p content in neurons under normal (i.e., nonstimulated conditions). Thus, we evaluated whether incubation with control sEVs modified the levels of miR-26a-5p in hippocampal neurons and we found that miR-26a-5p increases by ~30% in hippocampal neurons after the incubation with sEVs (right, Figure 3C). Together, these results suggest that the sEVs’ capacity to reduce dendritic complexity correlates with their content of miR-26a-5p, making it plausible that this molecule is carried by sEVs from astrocytes to neurons in order to regulate protein expression.

### 3.4. miR-26a-5p Mediates Aldo C-GFP sEVs-Induced Decrease of Dendritic Complexity

Next, we tested the hypothesis that increased levels of miR-26a-5p result in reduced dendritic complexity in neurons. To test this idea, we compared the dendritic morphology in hippocampal neurons magnetofected with mimic 26a-5p or with a scrambled control sequence (Figure 4A,B). We found that neurons magnetofected with mimic 26a-5p displayed a reduced dendritic complexity compared to scrambled, magnetofected neurons (Figure 4A,B). We also tested whether endogenous miR-26a-5p expressed by neurons plays a role in the regulation of dendritic complexity. However, magnetofection with antago 26a-5p did not alter dendritic morphology (Figure 4A,B). Therefore, increased but not endogenous levels of miR-26a-5p are active to regulate protein expression (Figure 3B) and dendritic complexity (Figure 4A,B) in hippocampal neurons.

Considering that magnetofection of mimic 26a-5p is sufficient to reduce dendritic complexity, (Figure 4A,B) then increased levels of miR-26-5p observed in Aldo C-GFP sEVs compared to GFP sEVs (Figure 1, Figure 2 and Figure 3) should contribute to reduced complexity of dendrites. Indeed, magnetofection of antago 26a-5p in hippocampal neurons prevented most of the reduction on the dendritic complexity induced by Aldo C-GFP sEVs (Figure 4C,D). Accordingly, magnetofection with the control scrambled sequence did not modify the response of neurons to Aldo C-GFP sEVs carrying the glial miR-26a-5p (Figure 4D).

## 4. Discussion

Here, we provided evidence for a role of astrocyte-derived sEVs in the regulation of neuronal morphology via control of miR-26a-5p activity in hippocampal neurons. Although many studies have focused on the functional impact of secreted, soluble astrocytic molecules on neurons, the role of astrocyte-derived sEVs has not yet been addressed in depth. Therefore, our data support the existence of a novel and complex level of intercellular astrocyte-to-neuron communication mediated by miRNAs in sEVs.

### 4.1. Astrocyte-Derived sEVs Contain Regulated miRNAs: A Potential New Role for Aldo C

Astrocytes can dynamically control the composition of sEVs by modifying their cargo contents in response to internal or external demands [8,47,48]. Here, we showed that Aldo C is present in astrocyte-derived sEVs and that the overexpression of Aldo C modulated the content of miR-26a-5p both in astrocyte homogenates and in the derived sEVs. 

We detected that overexpression of Aldo C induced a much larger and variable change of miR-26a-5p content in sEVs compared to homogenates. This could be partially explained by differences in the relative abundance of miR-26a-5p between sEVs and astrocyte homogenates, but also by the intrinsic complexity of sEVs in terms of loading control normalization [49,50,51]. For specific astrocyte-derived sEVs, there is not a clear consensus regarding normalization of miRNA levels. For example, one study used U6 small nuclear RNA to normalize miRNA content of homogenates, while rno-miR-23a was more suitable to normalize miRNA content of astrocyte-derived sEVs in response to ATP, TNFα, or IL-1β stimuli [8]. 

Our results showed that the total content of Aldo C correlates with increased miRNA-26a-5p levels, but importantly, that after inhibiting this miRNA activity, the morphological effect induced by sEVs were abolished. However, this does not exclude that other molecular players, such as different miRNAs, proteins, or lipids may also contribute to this mechanism. This emphasizes the complex nature of sEVs-mediated communication. 

It is unlikely that Aldo C overexpression led to increased miR-26a-5p by changing the metabolic flux of glial glycolysis because this catalytic step (aldohydrolysis of fructose 1,6-diphosphate to dihydroxyacetone phosphate (DHAP) and glyceraldehyde-3-phosphate (GAP)) is tightly regulated and close to thermodynamic equilibrium [52]. Indeed, it has been shown that after depleting 80% of aldolase B in tumor cells, the metabolic flow does not change significantly [53]. Furthermore, neurons are predominantly aerobic, which could make the Aldo C contribution to their overall metabolism less important than in other cell types [54]. Alternatively, revision of aldolases/Aldo C non-canonical functions, i.e., not associated to its function in glycolysis, may give clues regarding the upregulation of miR-26a-5p in astrocytes and sEVs [55,56,57,58]. Among them, Aldo C is able to activate the canonical Wnt pathway by forming a complex with Axin [59]. Interestingly, it has been shown that the activation of the Wnt signaling pathway through stimulation with a Wnt5a ligand triggers changes in the miRNA signature of hippocampal neurons [60]. Whether similar mechanisms operate in astrocytes remains as an open question. Here, the data point toward scenarios potentially related to non-canonical roles of Aldo C in the regulation sEVs, including their miRNA content.

### 4.2. A Role for sEVs in Transferring Glial miRNAs to Neurons and Their Impact on Neuronal Morphology

Our results suggested that increased miRNA-26a-5p levels in neurons via astrocyte-derived sEVs have a significant effect on neuronal morphology. It is well known that miRNA-26a-5p is endogenously expressed by neurons as well as astrocytes [35,61]. However, we found that a decrease in the endogenous miRNA-26a-5p activity by antago 26a-5p did not produce a noticeable effect in neurons. One interesting possibility to explain those results is that exogenously added miRNAs may reach target mRNAs more efficiently compared to the endogenous miRNAs. For example, it has been shown that sEVs could be destined to the recipient cell’s multivesicular bodies (MVBs) once they have been internalized, and that MVBs could be considered as signaling platforms in close proximity with the specialized RNA interference (RNAi) machinery [11,62]. Following the same reasoning, even if the miRNAs carried by astrocyte-derived sEVs may not be abundant compared to basal levels in neurons in absolute terms, they may still be effective in changing recipient cell’s function.

An interesting finding in this study, using bioinformatics, was the identification of a small cluster of glial miRNAs with the capacity to regulate the expression levels of molecular targets functionally related to neuronal development and neuronal morphology. This list of identified miRNAs and other miRNAs such as miR-132, miR379-410, miR-181a, miR-132/12 known to regulate of dendritic morphology [63,64,65,66] are evidence of a large capacity of miRNAs to regulate potentially several neuronal processes involved in neuronal development and morphology. Furthermore, evidence points to changes on regulation of neuronal morphology by miRNAs in pathological and/or neurological conditions (see later). Therefore, it would be interesting to perform quantitative transcriptomic analysis to compare the complete and precise content of miRNAs in sEVs from control astrocytes and from Aldo C-GFP astrocytes.

We found that miR-26a-5p has a key role in the control of the dendritic arborization. The evidence suggested that miR-26a-5p activity may have a compartmentalized response which would operate differently for the outgrowth of dendrites and axons. For instance, increased miR-26a-5p enhanced axon outgrowth in hippocampal neurons and axon regeneration in the peripheral nervous system [44,67]. Interestingly, this dual cell response is also observed after decreasing GSK3β (a validated miR-26a-5p target) activity in hippocampal neurons [68]. Additionally, miR-26a-5p is a direct regulator of Wnt5a, whose activity is critical for axonal outgrowth in cortical neurons, without compromising dendritic arborization [69,70]. We even validated that miR-26a-5p enhanced axon outgrowth in our model using staining of axons with βIII-tubulin (data not shown). Thus, it is tempting to speculate that endogenous neuronal miR-26a-5p and sEVs-transferred glial miR-26a-5p may have differential activity in axonal and dendritic compartments.

MiRNAs are known to operate in a complex fashion by controlling the expression of different genes that may impact on similar cellular functions [71]. In agreement with literature, we confirmed the control of miR-26a-5p over relevant targets such as MAP2 and GSK3β. Validated molecular targets of miR-26a-5p are key for the establishment of the extension and number of dendrites in hippocampal neurons. Accordingly, MAP2 down-regulation is known to induce loss of dendritic complexity and less elongation capacity of dendrites [45,72]. In contrast to this, increased expression of MAP2 is associated to larger dendritic length and complexity of cultured cerebellar neurons [73]. This effect on dendritic development is also compatible with another well validated target of miR-26a-5p such as brain derived neurotrophic factor (BDNF) [74]. Accordingly, decreased BDNF signaling is associated to poorer dendritic branching in developing hippocampal neurons, and conversely, activation of BDNF controlled pathways increases dendritic complexity [75,76]. Analogously, decreased GSK3β activity decreases dendritic length, while dendritic length increases after GSK3β over-expression [68]. Thus miR-26a-5p, when transferred via sEVs from astrocytes to neurons, controls several targets with high impact on the dendritic morphology, which is fully consistent with our results. It remains to be defined how miR-26a-5p impacts on mature neuronal morphology, including the dendritic tree, dendritic spines and thus, synaptic transmission.

### 4.3. Astrocyte-Derived sEVs in Neurological Conditions

The present work may uncover a novel link between stress-induced adaptations and their potential impact over dendritic structure. Increased Aldo C content in telencephalic astrocytes may have functional relevance in some mental disorders. Accordingly, Aldo C is up-regulated in the cortex of patients with schizophrenia, bipolar disease, and depression [77,78]. The Aldo C phosphorylation pattern is modified on the cerebrospinal fluid (CSF) of patients with major depressive disorder [79]. Furthermore, our group has shown that extracellular levels of Aldo C were highly up-regulated in a microsomal brain sub-fraction and in rat CSF after treatment with the antidepressant drug fluoxetine and in CSF and serum sEVs of rats exposed to stress by movement restriction [17,18,19]. Thus, we speculate that increased presence of Aldo C in astrocytes, as well as sEVs, may potentially reproduce the herein-described effects over dendritic structures of hippocampal neurons in pathological conditions.

The control of the dendritic arbor development is relevant as it may determine the computational capabilities and behavior in the adult brain [80]. Accordingly, abnormal dendritic development is seen in pathological states such as fragile-X mental retardation syndrome, Alzheimer’s disease, schizophrenia, and stress-induced adaptations [81,82,83,84]. Interestingly, it has been recently shown that inflammatory stimuli are able to modify the miRNA cargo of miR-125a-5p and miR-16-5p in glial sEVs to decrease the dendritic complexity of developing hippocampal neurons [8]. These examples reflect the new, exciting, and growing evidence for a role of exosome-like vesicles in mental disorders [85]. 

In the present study we examined the effects of astrocyte-derived sEVs on neurons during early developing stages, however, we may imagine a scenario in which sEVs impact the development of newborn neurons that will become continuously integrated to functional hippocampal circuits in adults. It is tempting to speculate that sEVs from the maternal blood stream may reach the embryo’s brain and modify dendritic development using the herein-described mechanism. This would make sense considering that hippocampal cells are extremely sensitive to stress-induced adaptations, which include dendritic atrophy [86]. Similar signaling, but in the opposite way, has been shown to occur in humans, where fetal-derived exosome-like vesicles reach the maternal blood stream [87]. Interestingly, growing evidence shows that exosome-like vesicles from astrocytic origin may reach the blood stream [19,88] and thus may have the capacity to signal to peripheral organs or body systems.

In summary, the present work suggests that astrocytes can regulate the dendritic development of neurons by modifying the miRNA cargo of their derived sEVs. This result raises novel questions regarding the precise elements controlling miRNA loading into astrocyte-derived sEVs and their impact on the function of developing as well as mature neurons.

## Figures and Tables

**Figure 1 cells-09-00930-f001:**
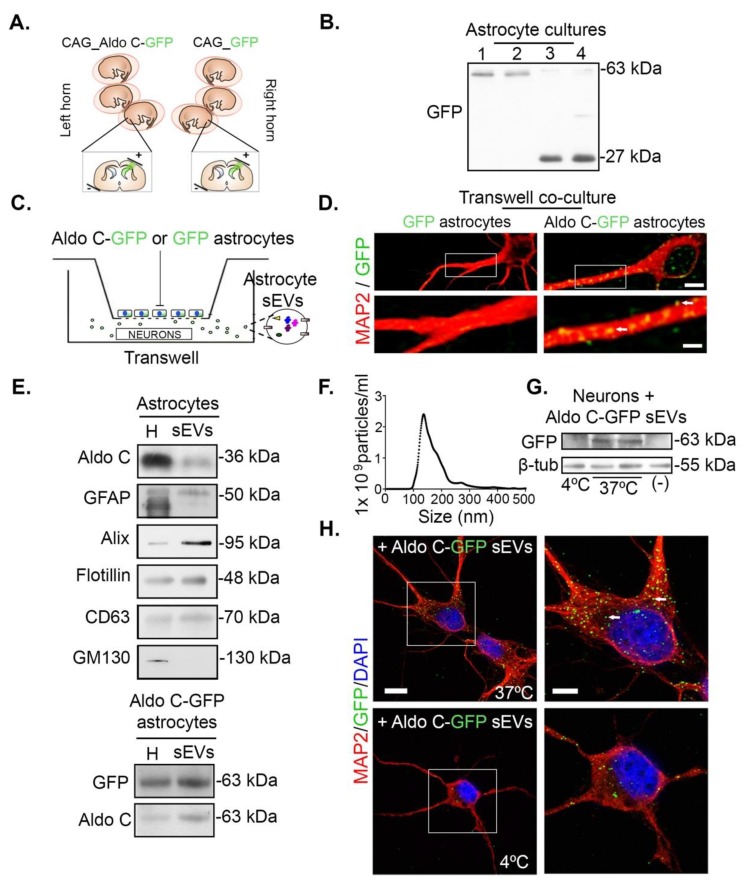
Astrocytes release small extracellular vesicles (sEVs) containing Aldo C-GFP, which are incorporated by hippocampal neurons. (**A**). Schematic drawing depicting in utero electroporation of cerebral cortices. Lateral ventricles of fetuses in the left horn were injected with Aldo C-GFP while GFP was injected in the right horn to apply the voltage pulse. Both constructs are stably inserted in the astrocyte’s genome and expressed under the control of the ubiquitous and strong promoter CAG. (**B**). Representative Western blots of cell lysates from astrocyte cell cultures expressing either Aldo C-GFP (lanes 1 and 2) or GFP (lanes 3 and 4). GFP antibody detects Aldo C-GFP (63 kDa) or GFP (27 kDa) bands in the corresponding condition. The same amount of protein was loaded in each lane and controlled by Coomassie staining (not shown). *N* = 4 independent experiments. (**C**). Schematic depicting the co-culture of astrocytes and neurons in the Transwell assay. (**D**). Confocal plane of neurons co-cultured either with GFP- or Aldo C-GFP-expressing astrocytes (GFP and Aldo C-GFP, respectively). Staining for MAP2 (red) and the GFP signal (green) are displayed. Somatodendritic compartments are positive for MAP2 (red). Neurons co-cultured with Astro Aldo C-GFP showed a high number of GFP+ puncta corresponding to Aldo C-GFP (green). GFP+ puncta are depicted by arrows in white. Scale Bar: 5 µm. Scale Bar insert: 2.5 µm. (**E**). Upper panel, representative Western blots from total homogenate (H) and sEV fraction derived from cultured astrocytes. The sEV markers CD63, Alix, and flotillin are detected in both samples. While the Golgi marker GM130 is detected only in homogenates, the astrocytic proteins Aldo C and GFAP are present in both samples. Lower panel, Western blot of homogenate (H) and isolated sEV fraction derived from Aldo C-GFP cultures. The recombinant protein Aldo C-GFP is detected at 63 kDA using either anti-GFP or anti-Aldo-C antibodies. (**F**). Nanoparticle tracking analysis of isolated sEVs (*N* = 4, sEVs obtained from control astrocytes). (**G**). Representative Western blots from hippocampal neurons (N) treated with Aldo C-GFP sEVs at 4 °C or 37 °C or vehicle as a negative control (-) during three hours at 37 °C. Recombinant Aldo C-GFP is found exclusively in homogenates from neurons incubated with Aldo C-GFP sEVs at 37 °C. β-actin is the loading control. (**H**). Representative confocal plane of hippocampal neurons treated with Aldo C-GFP sEVs at either 4 °C or 37 °C. Staining for MAP2 (red), GFP signal (green), and nuclei labelling with DAPI (blue) are displayed. GFP+ puncta are depicted by arrows in white. Scale Bar: 10 µm. Scale Bar insert: 5 µm.

**Figure 2 cells-09-00930-f002:**
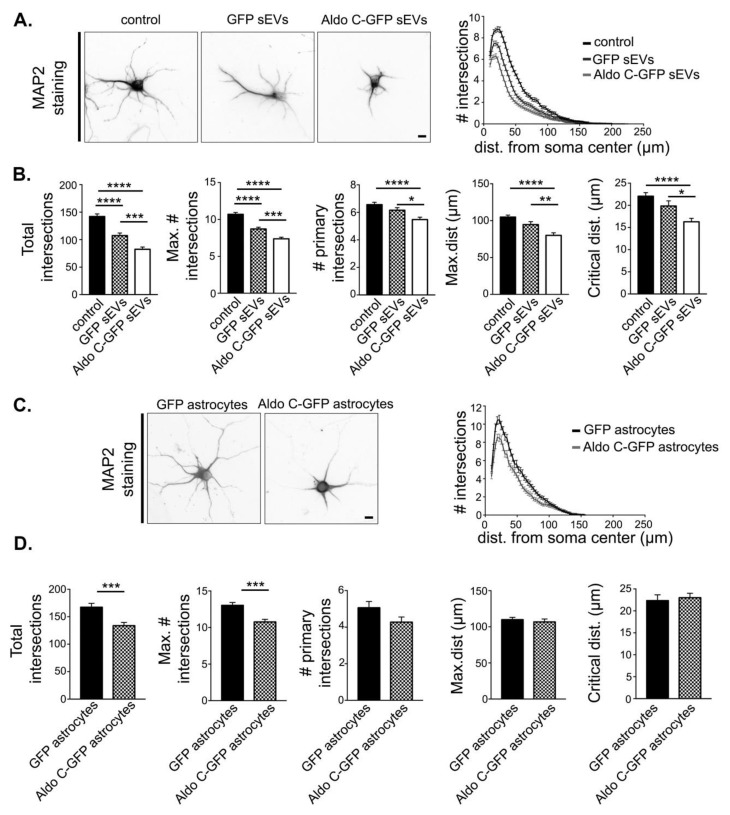
The sEVs derived from astrocytes decrease dendritic complexity of hippocampal neurons. (**A**). Left, representative inverted images of hippocampal neurons incubated with sEVs were stained with MAP2 for morphological analysis. Scale Bar: 10 µm. Right, Sholl analysis of neurons treated with vehicle only (control) or incubated with sEVs derived from GFP expressing astrocytes (GFP sEVs) or Aldo C-GFP sEVs. (**B**). Quantification of morphological parameters, namely total intersections (control 142 ± 5, *n* = 141; GFP sEVs 107 ± 5, *n* = 116; Aldo C-GFP sEVs 82 ± 4, *n* = 126), maximum number of intersections (max. # intersections) (control 10.7 ± 0.3; GFP sEVs 8.7 ± 0.2; Aldo C-GFP sEVs 7.4 ± 0.2), number of primary intersections (# primary intersections) (control 6.6 ± 0.2; GFP sEVs 6.2 ± 0.2; Aldo C-GFP sEVs 5.5 ± 0.2), maximum distance (max. dist.) (control 105 ± 3; GFP sEVs 95 ± 4; Aldo C-GFP sEVs 80 ± 3), and critical distance (control 22.1 ± 0.8; GFP sEVs 20 ± 1; Aldo C-GFP sEVs 16.3 ± 0.8) are displayed as mean ± standard error of *N* = 3 independent experiments. **** *p* < 0.0001, *** *p* < 0.001, ** *p* < 0.01, or * *p* < 0.05 using one-way ANOVA followed by Tukey’s multiple comparison test. (**C**). Left, representative inverted images of neurons co-cultured with either GFP or Aldo C-GFP astrocytes are displayed. Scale Bar: 10 µm. Right, Sholl analysis of neurons co-cultured with GFP- or Aldo C-GFP-expressing astrocytes (Astro GFP and Astro Aldo C-GFP, respectively). (**D**). Total intersections (Astro GFP 167 ± 7, *n* = 70; Astro Aldo C-GFP 134 ± 6, *n* = 70), maximum number of intersections (Astro GFP 13.0 ± 0.4; Astro Aldo C-GFP 10.8 ± 0.4), number of primary intersections (Astro GFP 5.1 ± 0.3; Astro Aldo C-GFP 4.3 ± 0.3), critical distance (Astro GFP 22 ± 1 µm; Astro Aldo C-GFP 23 ± 1; *p* = 0.6944), and maximal distance (Astro GFP 110 ± 3 µm; Astro Aldo C-GFP 107 ± 4 µm) are displayed as mean ± standard error of three independent experiments. **** *p* < 0.0001, *** *p* < 0.001, ** *p* < 0.01, or * *p* < 0.05 using the Welch’s *t*-test.

**Figure 3 cells-09-00930-f003:**
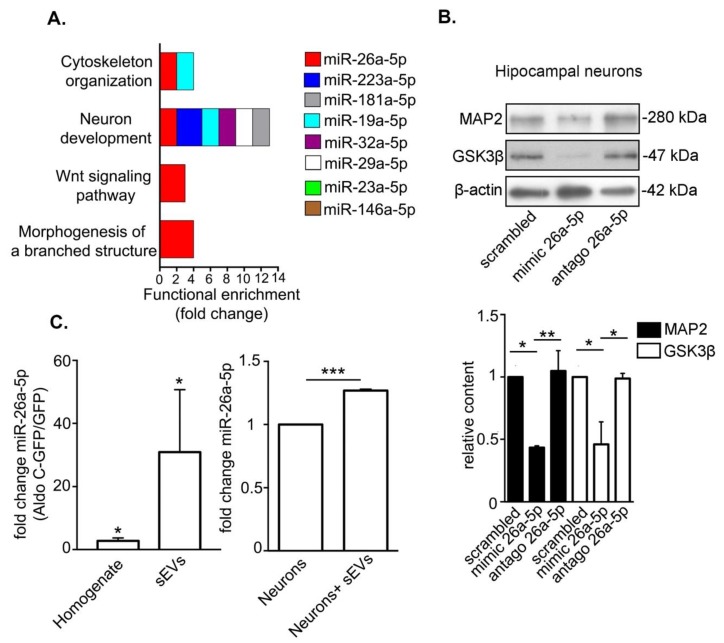
miR-26a-5p is predicted to control morphogenesis and its content is up-regulated in AldoC-GFP astrocytes and their released sEVs. (**A**). Functional enrichment analysis of predicted mRNAs targeted by selected miRNAs that have been reported in astrocytes. The size of the colored box is proportional to functional enrichment. Only functions with a *p* value < 0.05 are indicated after a Bonferroni test adjusted for binomial statistics. MiRNAs expression in astrocytes was confirmed from the following publications: miR-26a-5p, miR-23a-5p [35]; miR-223a-5p, miR-19a-5p, miR-32a-5p miR-146a-5p [33]; miR-181a-5p, miR-29a-5p [40]. (**B**). Upper panel, representative Western blots of 6 days in vitro (DIV) hippocampal neurons magnetofected with mimic 26a-5p, antago 26a-5p, and scrambled small oligonucleotides at 3 DIV. Lower panels, relative content of indicated proteins (normalized to the content of neurons magnetofected with scrambled). Relative content for MAP2 was as follows: mimic 26a-5p vs. scrambled, 0.43 ± 0.01 times; antago 26a-5p vs. scrambled, 1.0 ± 0.2 times. Relative content for GSK3β was as follows: mimic 26a-5p vs. scrambled 0.5 ± 0.2 times; antago 26a-5p vs. scrambled, 0.99 ± 0.04 times. Mean ± standard error of three independent experiments (*N* = 3) is indicated. *** *p* < 0.001; ** *p* < 0.01; or * *p* < 0.05 in one-way ANOVA Tukey’s multiple comparison test. (**C**). Left panel, RT-qPCR quantification of miR-26a-5p fold change in homogenates and sEVs from astrocytes that express Aldo C-GFP compared to GFP (control) and normalized to the U6 content. For homogenates, the fold change value was 2.8 ±0.9 times (*N* = 6). The value for sEVs was 31 ± 20 times (*N* = 7). Mean ± standard error is indicated. * *p* < 0.05. One-tailed Wilcoxon one-sample signed-rank test compared to a hypothetical value of 1.0. Right panel, quantification by qRT-PCR of miR-26a-5p fold change normalized to U6 of 6 DIV hippocampal neurons incubated with astrocyte sEVs for 3 h at 37 °C. Neuron + sEVs fold change value was 1.27 ± 0.01 times (*N* = 3). Mean ± standard error is indicated. *** *p* < 0.001 using one-tailed one sample *t*-test compared to a hypothetical value of 1.0.

**Figure 4 cells-09-00930-f004:**
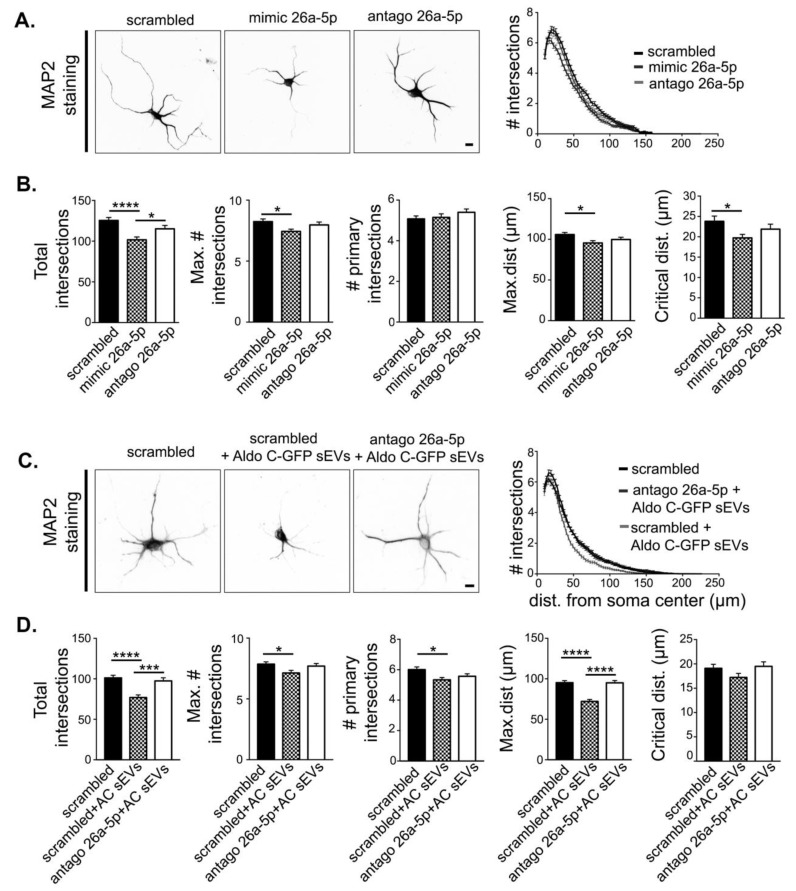
miR-26a mediates the Aldo C-GFP sEVs-induced decrease in the complexity of hippocampal dendrites. Hippocampal neurons were magnetofected either with scrambled, mimic 26a-5p, or antago 26a-5p sequences. (**A**). Left, representative inverted images of MAP2-stained neurons were used for the morphological quantification. Scale Bar: 10 µm. Right, Sholl analysis. (**B**). Total intersections (scrambled 125 ± 4, *n* = 119; mimic 26a-5p 102 ± 3, *n* = 120; antago 26a-5p 115 ± 4, *n* = 120), maximum number of intersections (max. # intersections) (scrambled 8.2 ± 0.2; mimic 26a-5p 7.4 ± 0.2; antago 26a-5p 8.0 ± 0.2), number of primary intersections (# primary intersections) (scrambled 5.1 ± 0.1; mimic 26a-5p 5.2 ± 0.2; antago 26a-5p 5.4 ± 0.2), maximum distance (max.dist.) (scrambled 106 ± 3; mimic 26a-5p 96 ± 3; antago 26a-5p 100 ± 3), and critical distance (scrambled 24 ± 1; mimic 26a-5p 19.7 ± 0.9; antago 26a-5p 22 ± 1) are displayed as mean ± standard error of *N* = 5 independent experiments. **** *p* < 0.0001, *** *p* < 0.001 or * *p* < 0.05 using one-way ANOVA followed by Tukey’s multiple comparison test. Hippocampal neurons were magnetofected with scrambled or antago 26a-5p and then treated with Aldo C-GFP sEVs. (**C**). Left, inverted images of MAP2-stained neurons were used for the morphological quantification. Scale Bar: 10 µm. Right, Sholl analysis. (**D**). Only for these depicted graphs Aldo C-GFP sEVs are denominated as AC sEVs. Total intersections (scrambled 101 ± 3, *n* = 217; scrambled+ Aldo C-GFP sEVs 77 ± 3, *n* = 173; antago 26a-5p + Aldo C-GFP sEVs 97 ± 4, *n* = 185), maximum number of intersections (scrambled 7.9 ± 0.2; scrambled+ Aldo C-GFP sEVs 7.0 ± 0.2; antago 26a-5p + Aldo C-GFP sEVs 7.7 ± 0.2), number of primary intersections (scrambled 6.0 ± 0.2; scrambled+ Aldo C-GFP sEVs 5.3 ± 0.2; antago 26a-5p + Aldo C-GFP sEVs 56 ± 0.2), maximum distance (scrambled 95± 2; scrambled+ Aldo C-GFP sEVs 72 ± 2; antago 26a-5p + Aldo C-GFP sEVs 95 ± 3), critical distance (scrambled 19.1 ± 0.8; scrambled+ Aldo C-GFP sEVs 17.2 ± 0.8; antago 26a-5p + Aldo C-GFP sEVs 19.5 ± 0.9) are displayed as mean ± standard error of *N* = 4 independent experiments. **** *p* < 0.0001, *** *p* < 0.001 or * *p* < 0.05 using one-way ANOVA followed by Tukey’s multiple comparison test.

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
