# Peer review of "Astrocyte-Derived Small Extracellular Vesicles Regulate Dendritic Complexity through miR-26a-5p Activity"

_cells, 2020, doi:10.3390/cells9040930_

Round 1

Reviewer 1 Report

Revision MS cells-734431

Astrocyte-derived small extracellular vesicles regulate dendritic complexity through miR-26a-5p activity.

Certainly an original work, very well presented and argued that provides new interpretations in the complex cellular communication astrocytes-neurons

I personally suggest to give more detailed results inherent to Nanoparticle tracking analysis (NTA). In particular, how many particles were used in each transwell assays. Also, even somehow described in Ref 19, a size profile of EVs populations might further complete the experimental details. Do also authors have the possibility to measure z-potential of their EVs ? .

These additional results are important in order to establish next reproducible experimental criteria

These data, sizing and possibly z-charge, might be inserted as supplemented materials.

Reviewer 2 Report

Luarte et al. have studied the effects of secreted sEVs from cultured astrocyts on the morphology of culturexd hippocampal neurons. They find that sEVs from astrocytes that overexpress AldoC produce a significant decrease in process branching, and provide evidence that the effect is due to the presence of miRNA-26a-5p in the exosomes. However, there are many molecules in sEVs. Even the control GFP-sEVs decrease process complexity in the cultured neurons. Thus, despite the fact that there is much more miRNA-26a-5p in the AldoC -GFP sEVs than in the GFP control sEVs, there are likely to be molecules in the control GFP sEVs that affect neuronal process complexity. This is an interesting finding, but further work needs to be done to strengthen the conclusions. 1. The authors should tell us by how much the electroporation of the AldoC vector increased astrocyte AldoC levels. They are working with cultured astrocytes in which AldoC is increased. 2. Are the numbers of sEVs secreted by the AldoC electroporated astrocytes similar to those of the GFP-electroporated or control astrocytes? An increase in sEVs, particularly the AldoC sEVs may explain some of the results. 3. Do neurons in these cultures contain miRNA-26a-5p? 4. What is the explanation for Figure 3C? There is a very large increase in miRNA-26a-5p in sEVs but little increase in neuronal miRNA-26a-5p levels. 5. The specificity of the effect of miRNA-26a-5p in neurons was assayed by magnetofecting the miRNA into neurons. The results are consistent with this hypothesis. A better experiment would have been to first delete or change the miRNA in the AldoC astrocytes so that the sEVs would not contain this miRNA and then look at the effects on neuronal morphology. 6. I think the authors mean on line 62: ”is not known whether neurons, which are in close proximity to astrocytes…”

Reviewer 3 Report

This study investigates the potential involvement of release of sEVs containing miR-26a-5p from astrocytes on neuronal dendritic complexity. While this is an interesting paper, there are a number of improvements that could be made to tighten up the study.

Major points

Fig. 1E shows that cultured astrocytes contain Aldo C (as expected) and that Aldo C is present in sEVs released from cultured astrocytes. The authors missed an opportunity to show how much Aldo C (native Aldo C vs Aldo C-GFP) is present in the astrocyte cultures prepared from the pups who received electroporation of the Aldo C-GFP plasmid in utero. Perhaps the authors could show the complete blots of the Aldo C-GFP astrocytes (the two bottom ones of Fig. 1E) in the supplementary data section. One would expect to see only the Aldo C-GFP (»63 kDa) band when using the anti-GFP antibody and both the native Aldo C (»36 kDa) and the Aldo C-GFP (»63 kDa) bands when using the anti-Aldo C antibody. This information becomes relevant when trying to interpret the data shown in Fig. 2 A and B.

Fig. 1G – Were the vehicle-treated neurons incubated at 37ºC or at 4ºC? This information should be included in the figure legend and/or the text.

Fig. 1 – In general, the quality of the Western blot data is not great. Perhaps it would have been better to optimize the dilution of each antibody rather than using 1:1,000 for all of them. In particular, the authors should discuss why there are apparent bands at »63 kDa in Fig. 1B (lanes 3 and 4). The pups used to prepare these astrocyte cultures were electroporated with the GFP plasmid in utero.

Fig. 2A and B – Is it the GFP or something else (perhaps miR-26a-5p) in normal astrocyte sEVs that is decreasing dendritic complexity? Did you try treating neurons with sEVs isolated from control astrocytes (not expressing GFP or Aldo C-GFP)? If so, how did the dendritic complexity compare with the other groups and particularly with the GFP group.

Fig. 3C – There is quite a bit of variability in the measurements of the left panel. For this reason, it would be best to include scattergrams and box plots to depict the variability in the measurements you present rather than the bar graph. This will give the reader much more information.

Fig. 3C right panel – Why did the authors chose to use sEVs from control astrocytes to examine if miR-26a-5p was increased in treated neurons? The authors should use sEVs from GFP and Aldo C-GFP expressing astrocytes for this study and this may be compared with the sEVs from control astrocytes.

Fig. 4 A and C – The data presented in the graphs plotting the # of intersections vs the distance from the soma are not as clear cut as those presented in figure 2.

Based on the conclusions of the authors, one would expect that the plot for the antago 26a-5p would be either similar to the scrambled or shifted to the right (Fig. 4A). Instead, the plot at distances greater than 50 µm from the soma is very similar to the mimic 26a-5p plot.

In Fig. 4C, the scrambled plot and the scrambled + Aldo C-GFP sEVs plot seem to overlap. This is very different from the effect of Aldo C-GFP sEVs shown in figure 2A. In addition, the antago 26a-5p + Aldo C-GFP sEVs plot is shifted to the left indicating that the antago is making complexity worse.

Do the authors care to comment about this?

Minor points

There are many typographical errors throughout the manuscript.

Here are some examples that should be fixed, but the authors should read the manuscript carefully for others.

            Line 62 I believe that the sentence should read “which are in close proximity to astrocytes”

            Line 247 and other places in the manuscript. This should be Welch’s t-test not Welsh’s t-test.

Line 196 – should be flotation not floatation

Line 175 – I doubt that the astrocyte cultures were pure. The reference to the method that is cited says that the astrocyte cultures are highly enriched. This should be changed in the text.

The figure legends are very wordy and difficult to read. Is it necessary to include the p values for every comparison? The methods state “The differences were considered statistically significant with p <0.05.”, therefore, it seems the legends could be simplified. The differences are either significant or not. A lower p value does not make a difference “more” significant.

Round 2

Reviewer 1 Report

Astrocyte-derived small extracellular vesicles regulate dendritic complexity through miR-26a-5p activity.

Dear Authors, I think you have justified the concerns I raised related to the above mentioned manuscript. In my opinion the manuscrip[t is suitable to be published in the Cells Journal.

Sincerely

Dr. Sergio Comincini, University of Pavia, Italy

Reviewer 2 Report

Authors have made quite a few revisions and improved the paper.  I recommend its publication. 

Reviewer 3 Report

The authors have addressed most of my concerns in the revised manuscript.  The exception to this is the large variability of the RT-qPCR measurements. The authors provide a table in the response to reviewers that shows the values for miR-26a-5p are much more similar than the values provided for the reference gene used for correction. This suggests that either U6 is the wrong reference gene to use or there are some technical inaccuracies in the procedure. Either way it makes it difficult to have a high degree of confidence in the RT-qPCR results.

Similarly, there is a tremendous variability in the RT-qPCR measurements shown in Supplemental Table 1 with some cases showing a decrease and others an increase in miR-26a-5p . The authors should either redo the samples shown in the supplemental table using another reference gene or at the very least they should show the actual values for the miR-26a-5p  and U6 for each sample (not just the fold change) as they did in the table provided in the response to reviewers. This would allow the readers to determine for themselves if they consider that these data are reliable.
